# RobustEmbed: Robust Sentence Embeddings Using Self-Supervised Contrastive Pre-Training

**Javad Rafiei Asl,[1] Eduardo Blanco,[2] Daniel Takabi[3]**
[1]Georgia State University, [2]University of Arizona, [3]Old Dominion University
jasl1@student.gsu.edu, eduardoblanco@arizona.edu, takabi@odu.edu

## Abstract

Pre-trained language models (PLMs) have demonstrated exceptional performance across a wide range of natural language processing tasks. The utilization of PLM-based sentence embeddings enables the generation of contextual representations that capture rich semantic information. However, despite their success with unseen samples, current PLM-based representations suffer from poor robustness in adversarial settings. In this paper, we propose RobustEmbed, a self-supervised sentence embedding framework that enhances both generalization and robustness in various text representation tasks and against a diverse set of adversarial attacks. By generating high-risk adversarial perturbations to promote higher invariance in the embedding space and leveraging the perturbation within a novel contrastive objective approach, RobustEmbed effectively learns high-quality sentence embeddings. Our extensive experiments validate the superiority of RobustEmbed over the state-of-the-art self-supervised representations in adversarial settings, while also showcasing relative improvements in seven semantic textual similarity (STS) tasks and six transfer tasks. Specifically, our framework achieves a significant reduction in attack success rate from 75.51% to 39.62% for the BERTAttack attack technique, along with enhancements of 1.20% and 0.40% in STS tasks and transfer tasks, respectively.

## 1 Introduction

Recent research has demonstrated the state-of-the-art performance of Pre-trained Language Models (PLMs) in learning contextual word embeddings (Devlin et al., 2019), leading to improved generalization in various Natural Language Processing (NLP) tasks (Yang et al., 2019; He et al., 2021; Ding et al., 2023). The focus of PLMs has extended to acquiring universal sentence embeddings, such as Universal Sentence Encoder (USE) (Cer et al., 2018) and Sentence-BERT (Reimers and Gurevych, 2019), which effectively capture the semantic representation of the input text. This representation learning facilitates feature generation for classification tasks and enhances large-scale semantic search (Neelakantan et al., 2022).

The assessment of PLM-based sentence representation relies on two crucial characteristics: generalization and robustness. While considerable research efforts have been dedicated to developing universal sentence embeddings using PLMs (Reimers and Gurevych, 2019; Zhang et al., 2020; Ni et al., 2022; Neelakantan et al., 2022; Wang et al., 2023; Bölücü et al., 2023), it is worth noting that despite their promising performance across various downstream classification tasks (Sun et al., 2019; Gao et al., 2021), demonstrating proficiency in generalization, these representations exhibit limitations in terms of robustness in adversarial settings and are vulnerable to diverse adversarial attacks (Nie et al., 2020; Wang et al., 2021). Existing research (Garg and Ramakrishnan, 2020; Wu et al., 2023; Hauser et al., 2023) highlights the poor robustness of these representations, such as BERT-based representations, which can be deceived by replacing a few words in the input sentence.

In this paper, we propose RobustEmbed, a robust sentence embedding framework that takes both of these essential characteristics into account. The core concept involves introducing a small adversarial perturbation to the input text and employing the contrastive objective (Chen et al., 2020) to learn high-quality sentence embeddings. RobustEmbed perturbs the embedding space rather than the raw text, which exhibits a positive correlation with generalization and promotes higher invariance. Our framework utilizes the original embedding along with the perturbed embedding as "positive pairs," while other sentence embeddings in the same mini-batch serve as "negatives." The contrastive objective identifies the positive pairs among the negatives. By incorporating norm-bounded adversarial

perturbation and contrastive objectives, our method enhances the robustness of similar sentences and disperses sentences with different semantics. This straightforward and efficient approach yields superior sentence embeddings in terms of both generalization and robustness benchmarks.

We conduct extensive experiments on a wide range of text representation and NLP tasks to verify the effectiveness of RobustEmbed including semantic textual similarity (STS) tasks (Conneau and Kiela, 2018), transfer tasks (Conneau and Kiela, 2018), and TextAttack (Morris et al., 2020). Two first series of experiments evaluate the quality of sentence embeddings on semantic similarity and natural language understanding tasks, while the last series assess the robustness of the framework against state-of-the-art adversarial attacks. RobustEmbed demonstrates significant improvements in robustness, reducing the attack success rate from 75.51% to 39.62% for the BERTAttack attack technique and achieving similar improvements against other adversarial attacks. Additionally, the framework achieves performance improvements of 1.20% and 0.40% on STS tasks and NLP transfer tasks, respectively, when employing the BERT$_{base}$ encoder.

**Contributions.** Our main contributions in this paper are summarized as follows:

- We introduce RobustEmbed, a novel self-supervised framework for sentence embeddings that generates robust representations capable of withstanding various adversarial attacks. Existing sentence embeddings are susceptible to such attacks, highlighting a vulnerability in their security. RobustEmbed fills this gap by employing high-risk perturbations within a novel contrastive learning approach.

- We conduct extensive experiments to demonstrate the efficacy of RobustEmbed across various text representation tasks and against state-of-the-art adversarial attacks. Empirical results confirm the high efficiency of our framework in terms of both generalization and robustness benchmarks.

- To facilitate further research in this important area, our source code is available in the RobustEmbed Repository

## 2 Related Work

The early work in text representations focused on applying the distributional hypothesis to predict words based on their context (Mikolov et al., 2013b,a). There are extensive studies on learning universal sentence embeddings using supervised and unsupervised approaches, such as Doc2vec (Le and Mikolov, 2014), SkipThought (Zhu et al., 2015), Universal Sentence Encoder (Cer et al., 2018), and Sentence-BERT (Reimers and Gurevych, 2019). More recently, self-supervised approaches have emerged, employing contrastive objectives to learn effective and robust text representations: SimCSE (Gao et al., 2021) introduced a minimal augmentation strategy to predict the input sentence by applying two different dropout masks. The ConSERT model (Yan et al., 2021) utilized four distinct data augmentation techniques to generate diverse views for the purpose of executing a contrastive objective: adversarial attacks, token shuffling, cut-off, and dropout. Qiu et al. (2021) introduced two adversarial training methods, CARL and RAR, to strengthen the ML model's defense against gradient-based adversarial attacks. CARL aims to acquire a resilient representation at the sentence level, whereas RAR focuses on enhancing the robustness of individual word representations. Rima et al. (2022) proposed adversarial training with contrastive learning for training natural language processing models. It involves applying linear perturbations to input embeddings and leveraging contrastive learning to minimize the distance between original and perturbed representations. Pan et al. (2022) presents a straightforward approach for regularizing transformer-based encoders during the fine-tuning step. The model achieves noise-invariant representations by generating adversarial examples perturbing word embeddings and leveraging contrastive learning.

In comparison to several existing contrastive adversarial learning approaches in the text representation area (Yan et al., 2021; Meng et al., 2022; Qiu et al., 2021; Li et al., 2023; Rima et al., 2022; Pan et al., 2022), our framework stands out by generating more efficient high-risk iterative perturbations in the embedding space. Furthermore, our framework leverages a more powerful contrastive objective approach, leading to high-quality text representations that demonstrate enhanced generalization and robustness properties. Empirical results substantiate the superiority of our approach across

various generalization and robustness benchmarks.

# 3 Background

In this section, we present an overview of the recent progress in adversarial perturbation generation and self-supervised contrastive learning.

## 3.1 Adversarial Perturbation Generation

Adversarial perturbation involves adding maliciously crafted perturbations to benign data, which can deceive Machine Learning (ML) models, including deep learning methods (Goodfellow et al., 2015). These perturbations are designed to be imperceptible to humans but can cause the model to make incorrect predictions (Metzen et al., 2017). Adversarial training, which involves incorporating adversarial perturbations during the model training process, has been shown to enhance the model's robustness against adversarial attacks (Madry et al., 2018; Shafahi et al., 2020; Xu et al., 2020; Wang et al., 2019b). While various perturbation generation techniques have contributed to machine vision (Chakraborty et al., 2021), the progress of these techniques in the NLP domain has been at a slower pace due to the discrete nature of text (Jin et al., 2020). In recent years, instead of directly applying adversarial perturbations to raw text, a few studies have focused on perturbing the embedding space (Wang et al., 2019a; Dong et al., 2021). However, these methods still face challenges in terms of generalization, as they may not be applicable to any ML model and NLP tasks. Utilized within our framework, a more generalized approach for generating high-risk adversarial perturbations involves applying a small noise $\delta$ within a norm ball to the embedding space, aiming to maximize the adversarial loss:

$$\arg \max_{||\boldsymbol{\delta}|| \leq \epsilon} L(f_\theta(X + \boldsymbol{\delta}), y), \qquad (1)$$

where $f_\theta(.)$ denotes an ML model parameterized with $X$ as the sub-word embeddings, and $y$ is the truth label. Various gradient-based algorithms have been proposed to address this optimization problem. We employ a practical combination of the Fast Gradient Sign Method (FGSM) (Goodfellow et al., 2015) and the Projected Gradient Descent (PGD) technique (Madry et al., 2018) to generate adversarial perturbations that represent worst-case examples.

## 3.2 Contrastive Learning Based Representation

The objective of contrastive learning is to acquire effective low-dimensional representations by bringing semantically similar samples closer and pushing dissimilar ones further apart (Hadsell et al., 2006). Self-supervised contrastive learning has demonstrated promising results in data representation across domains such as machine vision (Chen et al., 2020), natural language processing (Gao et al., 2021; Neelakantan et al., 2022), and speech recognition (Lodagala et al., 2023). Our framework adopts the contrastive learning concept proposed by Chen et al. (2020) to generate high-quality representations. Let $\{(x_i, x_i^+)\}_{i=1}^N$ denote a set of $N$ positive pairs, where $x_i$ and $x_i^+$ are semantically correlated and $(z_i, z_i^+)$ represents the corresponding embedding vectors for the positive pair $(x_i, x_i^+)$. We define $z_i$'s positive set as $\{x_i^{pos}\} = z_i^+$, while the negative set $\{x_i^{neg}\}$ as the set of other positive pairs. Then, the contrastive training objective can be defined as follows:

$$\mathcal{L}_{con,\theta}(x_i, \{x_i^{pos}\}, \{x_i^{neg}\}) = \qquad (2)$$
$$-\log\left(\frac{\sum_{\{x_i^{pos}\}} \exp(sim(z_i, \{x_i^{pos}\})/\tau)}{\sum_{\{x_i^{pos}, x_i^{neg}\}} \exp(sim(z_i, \{x_i^{pos}, x_i^{neg}\})/\tau)}\right),$$

where $\tau$ denotes a temperature hyperparameter and $sim(u, v) = \frac{u^\top v}{||u||.||v||}$ is the cosine similarity between two representation vectors. The standard objective function only contains a single sample in the positive set. The total loss is computed over all positive pairs within a mini-batch.

# 4 The Proposed Adversarial Self-supervised Contrastive Learning

We introduce RobustEmbed, a simple yet effective approach for generating universal text representations through adversarial training of a self-supervised contrastive learning model. Given a PLM $f_\theta(.)$ as the encoder and a large unsupervised dataset $\mathcal{D}$, RobustEmbed aims to pre-train $f_\theta(.)$ on $\mathcal{D}$ to enhance the efficiency of sentence embeddings across diverse NLP tasks (improved generalization) and increase resilience against various adversarial attacks (enhanced robustness). Algorithm 1 demonstrates our framework's approach to generating a norm-bounded perturbation using an iterative process, confusing the $f_\theta(.)$ model by treating the perturbed embeddings as different instances.

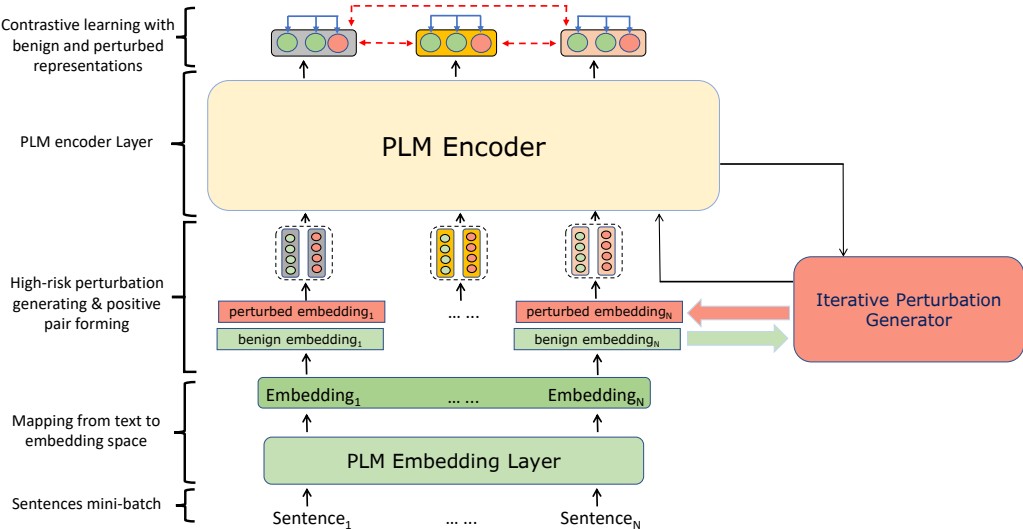

Figure 1: The general architecture of the RobustEmbed framework. In contrastive learning step, a blue arrow indicate gathering positive pairs together, and a red arrow refers to keeping distance among negative pairs

Our framework then employs a contrastive learning approach to maximize the similarity between the embedding of an input instance and the adversarial embedding of its positive pair. Moreover, Figure 1 provides an overview of our RobustEmbed framework, which aims to achieve adversarial robustness in representations. The framework involves an iterative collaboration between the perturbation generator and the $f_\theta(.)$ model to generate high-risk perturbations for adversarial contrastive learning during the final training step. The subsequent sections delve into the main components of our framework and provide a detailed analysis of the training objective.

### 4.1 Perturbation Generation

As the primary step, RobustEmbed aims to generate small perturbations that fool the ML model, leading to incorrect predictions, while remaining nearly imperceptible to humans. The framework uses an approach based on combination of the PGD and FGSM algorithms to generate a perturbation that maximizes the self-supervised contrastive loss, facilitating discrimination between various instances. RobustEmbed employs multiple iterations of this combination, specifically T-step FGSM and K-step PGD, to meticulously reinforce invariance within the embedding space, ultimately resulting in enhanced generalization and robustness.

In particular, considering the PLM-based encoder $f_\theta(.)$ and an input sentence $x$, RobustEmbed passes the sentence to the $f_\theta(.)$ model twice: by

---

**Algorithm 1: RobustEmbed Algorithm**

**Input:** Epoch number $E$, PLM Encoder $\mathbf{f_\theta}$, dataset of raw sentences $\mathcal{D} = \{x_i\}_{i=1}^N$, perturbation $\boldsymbol{\delta}$, dropout masks $m_1$ and $m_2$, perturbation bound $\epsilon$, step sizes $\alpha$ and $\beta$, learning rate $\eta$, perturbation modulator $\lambda$, regularization parameter $\gamma$, perturbation generation iterators $K$ and $T$, contrastive learning objective $\mathcal{L}_{con,\theta}$ (eq. 2)

**Output:** Robust Sentence Representation

**for** epoch $= 1, ..., E$ **do**
  **for** *minibatch* B $\subset \mathcal{D}$ **do**
    $\boldsymbol{\delta^1} \sim \mathcal{N}(0, \sigma^2 I)$
    $\boldsymbol{X} = \mathbf{f_\theta}.\text{embedding}(B, m_1)$
    $\boldsymbol{X^+} = \mathbf{f_\theta}.\text{embedding}(B, m_2)$
    **for** $t = 1, ..., max(K, T)$ **do**
      $g(\boldsymbol{\delta^t}) = \nabla_\delta \mathcal{L}_{con,\theta}(\boldsymbol{X} + \boldsymbol{\delta^t}, \{\boldsymbol{X^+}\})$
      **if** $t \leq K$ **then**
        $\boldsymbol{\delta}_{pgd}^{t+1} = \Pi_{\|\boldsymbol{\delta}\|_P \leq \epsilon}(\boldsymbol{\delta^t} + \alpha g(\boldsymbol{\delta^t})/\|g(\boldsymbol{\delta^t})\|_P)$
      **end**
      **if** $t \leq T$ **then**
        $\boldsymbol{\delta}_{fgsm}^{t+1} = \Pi_{\|\boldsymbol{\delta}\|_P \leq \epsilon}(\boldsymbol{\delta^t} + \beta \text{sign}(g(\boldsymbol{\delta^t})))$
      **end**
    **end**
    $\boldsymbol{\delta}_f = \lambda \boldsymbol{\delta}_{pgd}^K + (1 - \lambda)\boldsymbol{\delta}_{fgsm}^T$
    $\mathcal{L}_{RobustEmbed, \theta} := \mathcal{L}_{con,\theta}(\boldsymbol{X}, \{\boldsymbol{X^+}, \boldsymbol{X} + \boldsymbol{\delta}_f\})$
    $\mathcal{L}_{total} := \mathcal{L}_{RobustEmbed, \theta} + \gamma \mathcal{L}_{con,\theta}(\boldsymbol{X} + \boldsymbol{\delta}_f, \{\boldsymbol{X^+}\})$
    $\theta = \theta - \eta \nabla_\theta \mathcal{L}_{total}$
  **end**
**end**

---

applying the standard dropout twice, two different embeddings of $(X, X^+)$ are obtained as "positive pairs" (Gao et al., 2021). The framework takes the following steps to update the perturbation separately for the PGD and FGSM in iteration $k + 1$

and $t + 1$ respectively:

$$\boldsymbol{\delta}_{pgd}^{k+1} = \Pi_{\|\boldsymbol{\delta}\|_P \leq \epsilon}(\boldsymbol{\delta}^k + \alpha g(\boldsymbol{\delta}^k)/\|g(\boldsymbol{\delta}^k)\|_P), \quad (3)$$

$$\boldsymbol{\delta}_{fgsm}^{t+1} = \Pi_{\|\boldsymbol{\delta}\|_P \leq \epsilon}(\boldsymbol{\delta}^t + \beta \text{sign}(g(\boldsymbol{\delta}^t))), \quad (4)$$

where $g(\boldsymbol{\delta}^n) = \nabla_{\boldsymbol{\delta}} \mathcal{L}_{con,\theta}(\boldsymbol{X} + \boldsymbol{\delta}^n, \{\boldsymbol{X}^+\})$ with $n = t \text{ or } k$ is the gradient of the contrastive learning loss with respect to $\boldsymbol{\delta}$. The perturbation is generated by the $\ell_\infty$ norm-ball around the input embedding with radius $\epsilon$, and $\Pi$ projects the perturbation onto the $\epsilon$-ball. Further, $\alpha$ and $\beta$ are the step sizes of the attacks and $\text{sign}(.)$ returns the sign of the vector. Essentially, T-step FGSM and K-step PGD are mathematically equivalent when $P$ is either 2 or $\infty$. Their primary distinctions lie in the number of iterations (i.e., T and K) and the step size of the attack (i.e., $\alpha$ and $\beta$ ) used to modify the input perturbation, ultimately generating a unique high-level perturbation. The final perturbation can be obtained through the combination of T-step FGSM and K-step PGD:

$$\boldsymbol{\delta}_{final} = \lambda \boldsymbol{\delta}_{pgd}^K + (1 - \lambda)\boldsymbol{\delta}_{fgsm}^T, \quad (5)$$

where $0 \leq \lambda \leq 1$ modulates the relative significance of each separate perturbation in the generation of the final perturbation.

## 4.2   Robust Contrastive Learning

To achieve robust representation through self-supervised contrastive learning, adversarial learning objective, which follows a min-max formulation to minimize the maximum risk for any perturbation $\boldsymbol{\delta}$ (Madry et al., 2018), could be defined as follows:

$$\arg\min_\theta \mathbb{E}_{(x) \sim \mathcal{D}}[\max_{\|\boldsymbol{\delta}\| \leq \epsilon} \mathcal{L}_{con,\theta}(\boldsymbol{X} + \boldsymbol{\delta}, \{\boldsymbol{X}^+\})], \quad (6)$$

where $\boldsymbol{X} + \boldsymbol{\delta}$ is the adversarial embedding generated by the iterative gradient-based perturbation generation (eq. 5). Our framework utilizes adversarial examples generated in the embedding space, rather than using the original raw text, resulting in an ultimate pre-trained model that is robust against m-way instance-wise adversarial attacks. The framework employs the contrastive learning objective to maximize the similarity between clean examples and their adversarial perturbation by incorporating the adversarial example as the additional element in the positive set:

$$\mathcal{L}_{RobustEmbed, \theta} := \mathcal{L}_{con,\theta}(x, \{x^{pos}, x^{adv}\}), \quad (7)$$

$$\mathcal{L}_{total} := \mathcal{L}_{RobustEmbed, \theta} + \gamma \mathcal{L}_{con,\theta}(x^{adv}, \{x^{pos}\}), \quad (8)$$

where $x^{adv}$ represents the adversarial perturbation of the input sample $x$ in the embedding space, and $\gamma$ denotes a regularization parameter. The first part of the total contrastive loss (eq. 8) aims to optimize the similarity between the input sample $x$, its positive pair, and its adversarial perturbation, while the second part serves to regularize the loss by encouraging the convergence of the adversarial perturbation and the positive pair of $x$.

## 5   Evaluation and Experimental Results

This section presents a comprehensive set of experiments aimed at validating the effectiveness of our proposed framework in terms of generalization and robustness metrics. In the first two series of experiments, we investigate the performance of our framework on seven semantic textual similarity (STS) tasks and six transfer tasks within the SentEval framework[1] to assess the generalization capability of our framework in generating efficient sentence embeddings. In the final series of experiments, we measure the resilience of the embeddings against five state-of-the-art adversarial attacks to assess the robustness capability of our framework in generating robust text representation. Appendices A and B provide training details and ablation studies that illustrate the effects of hyperparameter tuning.

### 5.1   Semantic Textual Similarity (STS) Tasks

We evaluate our framework on a set of seven semantic textual similarity (STS) tasks, which include STS 2012–2016 (Agirre et al., 2012, 2013, 2014, 2015, 2016), STS Benchmark (Cer et al., 2017), and SICK-Relatedness (Marelli et al., 2014). In our experiments, we solely utilize fixed sentence embeddings without any training datasets or regressors. To benchmark our framework's performance, we compare it against various unsupervised sentence embedding approaches, including: 1) baseline methods such as GloVe (Pennington et al., 2014) and average BERT or RoBERTa embeddings; 2) post-processing methods like BERT-flow (Li et al., 2020a) and BERT-whitening (Su et al., 2021); and 3) state-of-the-art methods such as SimCSE (Gao et al., 2021), ConSERT (Yan et al., 2021), USCAL (Miao et al., 2021), and ATCL (Rima et al., 2022). We validate the findings of the SimCSE, ConSERT, and USCAL frameworks

---

[1] https://github.com/facebookresearch/SentEval

| Model | STS12 | STS13 | STS14 | STS15 | STS16 | STS-B | SICK-R | Avg. |
|---|---|---|---|---|---|---|---|---|
| GloVe embeddings (avg.) $^\heartsuit$ | 55.14 | 70.66 | 59.73 | 68.25 | 63.66 | 58.02 | 53.76 | 61.32 |
| BERT$_{base}$ (first-last avg.) $\clubsuit$ | 39.70 | 59.38 | 49.67 | 66.03 | 66.19 | 53.87 | 62.06 | 56.70 |
| BERT$_{base}$-flow $\clubsuit$ | 58.40 | 67.10 | 60.85 | 75.16 | 71.22 | 68.66 | 64.47 | 66.55 |
| BERT$_{base}$-whitening $\clubsuit$ | 57.83 | 66.90 | 60.90 | 75.08 | 71.31 | 68.24 | 63.73 | 66.28 |
| ConSERT-BERT$_{base}$ | 64.56 | 78.55 | 69.16 | 79.74 | 76.00 | 73.91 | 67.35 | 72.75 |
| ATCL-BERT$_{base}$ | 67.14 | 80.86 | 71.73 | 79.50 | 76.72 | 79.31 | 70.49 | 75.11 |
| SimCSE-BERT$_{base}$ | 68.66 | 81.73 | 72.04 | 80.53 | **78.09** | 79.94 | 71.42 | 76.06 |
| USCAL-BERT$_{base}$ | 69.30 | 80.85 | 72.19 | 81.04 | 77.52 | 81.28 | 71.98 | 76.31 |
| ⋆RobustEmbed-BERT$_{base}$ | **70.52** | **82.13** | **73.56** | **82.38** | 77.72 | **82.97** | **73.24** | **77.51** |
| RoBERTa$_{base}$-whitening $^\square$ | 46.99 | 63.24 | 57.23 | 71.36 | 68.99 | 61.36 | 62.91 | 61.73 |
| ConSERT-RoBERTa$_{base}$ | 66.90 | 79.31 | 70.33 | 80.57 | 77.95 | 81.42 | 68.16 | 74.95 |
| SimCSE-RoBERTa$_{base}$ | 68.75 | 80.81 | 71.19 | 81.79 | 79.35 | 82.62 | 69.56 | 76.30 |
| USCAL-RoBERTa$_{base}$ | 69.28 | 81.15 | 72.81 | 81.47 | **80.55** | 83.34 | 70.94 | 77.08 |
| ⋆RobustEmbed-RoBERTa$_{base}$ | **69.71** | **81.77** | **73.34** | **81.98** | 79.74 | **83.70** | **71.10** | **77.33** |
| USCAL-RoBERTa$_{large}$ | 68.70 | **81.84** | 74.26 | 82.52 | **80.01** | 83.14 | 76.30 | 78.11 |
| ⋆RobustEmbed-RoBERTa$_{large}$ | **68.92** | 81.53 | **74.35** | **82.91** | 79.98 | **83.93** | **76.93** | **78.36** |

Table 1: Semantic Similarity performance on STS tasks (Spearman's correlation, "all" setting) for sentence embedding models. We emphasize the top-performing numbers among models that share the same pre-trained encoder. $\heartsuit$: results from (Reimers and Gurevych, 2019); $\clubsuit$: results from (Gao et al., 2021); All remaining results have been reproduced and reevaluated by our team. The ⋆ symbol shows our framework.

by reproducing their results using our own configuration for BERT and RoBERTa encoders. The results presented in Table 1 demonstrate the superior performance of our RobustEmbed framework compared to various sentence embedding methods across most of the semantic textual similarity tasks. Our framework achieves the highest averaged Spearman's correlation among state-of-the-art approaches. Specifically, when using the BERT encoder, our framework outperforms the second-best embedding method, USCAL, by a margin of 1.20%. Additionally, RobustEmbed achieves the highest score in the majority of individual STS tasks (6 out of 7) compared to other embedding methods and performs comparably to the SimCSE method on the STS16 task. For the RoBERTa encoder, both the base version and the large version, RobustEmbed outperforms the state-of-the-art embeddings in five out of seven STS tasks and achieves the highest averaged Spearman's correlation.

## 5.2 Transfer Tasks

This experiment leverages transfer tasks to evaluate the performance of our framework, RobustEmbed, on diverse text classification tasks, including sentiment analysis and paraphrase identification. Our assessment encompasses six transfer tasks: CR (Hu and Liu, 2004), SUBJ (Pang and Lee, 2004), MPQA (Wiebe et al., 2005), SST2 (Socher et al., 2013), and MRPC (Dolan and Brockett, 2005),

with detailed information provided in Appendix E. We adhere to the standard methodology described in Conneau and Kiela (2018) and train a logistic regression classifier on top of the fixed sentence embeddings for our experimental procedure. We replicated the SimCSE, ConSERT, and USCAL frameworks using our configuration for both BERT and RoBERTa encoders. The results presented in Table 2 indicate that our framework demonstrates superior performance in terms of average accuracy when compared to other sentence embedding methods. Specifically, when utilizing the BERT encoder, our framework outperforms the second-best embedding method by a margin of 0.40%. Moreover, RobustEmbed achieves the highest score in four out of six text classification tasks. The similar interpretation of the BERT encoder are also maintained for the RoBERTa encoder, including both the base version and the large version.

## 5.3 Adversarial Attacks

In this section, we evaluate the robustness of our sentence embedding framework against various adversarial attacks, comparing it with two state-of-the-art sentence embedding models: SimSCE (Gao et al., 2021) and USCAL (Miao et al., 2021). Our evaluation involves fine-tuning a BERT-based PLM using different embedding approaches on seven text classification and natural language inference tasks, namely MRPC (Dolan and Brockett,

| Model | MR | CR | SUBJ | MPQA | SST2 | MRPC | Avg. |
|---|---|---|---|---|---|---|---|
| GloVe embeddings (avg.) ♣ | 77.25 | 78.30 | 91.17 | 87.85 | 80.18 | 72.87 | 81.27 |
| Skip-thought ♡ | 76.50 | 80.10 | 93.60 | 87.10 | 82.00 | 73.00 | 82.05 |
| BERT-`[CLS]` embedding ♣ | 78.68 | 84.85 | 94.21 | 88.23 | 84.13 | 71.13 | 83.54 |
| ConSERT-BERT$_{base}$ | 79.52 | 87.05 | 94.32 | 88.47 | 85.46 | 72.54 | 84.56 |
| SimCSE-BERT$_{base}$ | 81.29 | 86.94 | 94.72 | 89.49 | **86.70** | 75.13 | 85.71 |
| USCAL-BERT$_{base}$ | 81.54 | 87.12 | 95.24 | 89.34 | 85.71 | 75.84 | 85.80 |
| ⋆RobustEmbed-BERT$_{base}$ | **81.94** | **87.45** | 95.04 | **89.88** | 86.47 | **76.40** | **86.20** |
| SimCSE-RoBERTa$_{base}$ | 81.15 | 87.15 | 92.38 | 86.79 | **86.24** | 75.49 | 84.87 |
| USCAL-RoBERTa$_{base}$ | **82.15** | 87.22 | 92.76 | 87.74 | 84.39 | 76.20 | 85.08 |
| ⋆RobustEmbed-RoBERTa$_{base}$ | 81.49 | **87.54** | **93.37** | **87.95** | 84.63 | **76.62** | **85.27** |
| USCAL-RoBERTa$_{large}$ | **82.84** | 87.97 | 93.12 | 88.48 | **86.28** | 76.41 | 85.85 |
| ⋆RobustEmbed-RoBERTa$_{large}$ | 82.38 | **88.27** | **93.91** | **88.79** | 86.01 | **77.11** | **86.08** |

Table 2: Results of transfer tasks for different sentence embedding models. ♣: results from (Reimers and Gurevych, 2019); ♡: results from (Zhang et al., 2020); We emphasize the top-performing numbers among models that share the same pre-trained encoder. All remaining results have been reproduced and reevaluated by our team. The ⋆ symbol shows our framework.

| Adversarial Attack | Model | IMDB | MR | SST2 | YELP | MRPC | SNLI | MNLI-Mismatched | Avg. |
|---|---|---|---|---|---|---|---|---|---|
| | SimCSE-BERT$_{base}$ | 75.32 | 65.53 | 71.49 | 79.67 | 80.07 | 72.65 | 68.54 | 72.61 |
| TextFooler | USCAL-BERT$_{base}$ | 61.94 | 48.71 | 55.38 | 62.30 | 60.18 | 54.82 | 53.74 | 56.72 |
| | RobustEmbed-BERT$_{base}$ | **40.55** | **32.69** | **36.17** | **44.25** | **38.88** | **37.61** | **35.63** | **37.97** |
| | SimCSE-BERT$_{base}$ | 52.21 | 42.04 | 49.67 | 56.19 | 56.73 | 45.39 | 40.16 | 48.91 |
| TextBugger | USCAL-BERT$_{base}$ | 39.16 | 27.37 | 31.90 | 41.25 | 37.86 | 30.79 | 25.45 | 33.40 |
| | RobustEmbed-BERT$_{base}$ | **23.70** | **18.03** | **20.24** | **28.58** | **20.89** | **19.07** | **16.33** | **20.98** |
| | SimCSE-BERT$_{base}$ | 64.41 | 55.73 | 60.48 | 67.54 | 68.15 | 56.09 | 52.58 | 60.71 |
| PWWS | USCAL-BERT$_{base}$ | 51.95 | 40.67 | 45.29 | 52.30 | 46.86 | 50.92 | 39.37 | 46.77 |
| | RobustEmbed-BERT$_{base}$ | **33.63** | **28.15** | **30.56** | **29.94** | **25.51** | **27.16** | **28.49** | **29.06** |
| | SimCSE-BERT$_{base}$ | 73.50 | 61.83 | 68.27 | 75.15 | 77.84 | 69.06 | 65.43 | 70.15 |
| BAE | USCAL-BERT$_{base}$ | 58.57 | 46.19 | 51.72 | 59.49 | 58.38 | 50.90 | 51.16 | 53.77 |
| | RobustEmbed-BERT$_{base}$ | **37.35** | **29.82** | **32.08** | **41.66** | **36.45** | **34.17** | **31.98** | **34.79** |
| | SimCSE-BERT$_{base}$ | 78.42 | 66.94 | 73.59 | 80.87 | 82.16 | 74.35 | 72.22 | 75.51 |
| BERTAttack | USCAL-BERT$_{base}$ | 63.23 | 51.08 | 57.73 | 63.96 | 63.05 | 55.41 | 55.86 | 58.62 |
| | RobustEmbed-BERT$_{base}$ | **42.30** | **34.76** | **38.81** | **45.15** | **39.97** | **39.08** | **37.24** | **39.62** |

Table 3: Attack success rates of various adversarial attacks applied to three sentence embeddings (SimCSE-BERT, USCAL-BERT, and RobustEmbed-BERT) across five text classification and two natural language inference tasks.

2005), YELP (Zhang et al., 2015), IMDb (Maas et al., 2011), Movie Reviews (MR) (Pang and Lee, 2005a), SST2 (Socher et al., 2013), Standford NLI (SNLI) (Bowman et al., 2015), and Multi-NLI (MNLI) (Williams et al., 2018). Detailed information regarding these tasks can be found in Appendix E. To assess the robustness of the fine-tuned models, we perform adversarial attacks using the TextAttack framework (Morris et al., 2020) to investigate the impact of five efficient adversarial attack techniques: TextBugger (Li et al., 2019), PWWS (Ren et al., 2019), TextFooler (Jin et al., 2020), BAE (Garg and Ramakrishnan, 2020), and

BERTAttack (Li et al., 2020b). To acquire a more comprehensive insight into the functionality of these attacks, we provide more details in Appendix F. It should be noted that adaptive attacks cannot generate adversarial attacks using the main algorithm of our framework, as it operates exclusively in the embedding space while the input instances of sentence embeddings are raw text. To ensure statistical validity, each experiment was conducted five times, each time using 1000 adversarial attack samples; the reported results shown in this section are the average results of five iterations.

Table 3 presents the attack success rates of five

adversarial attack techniques on three sentence embeddings, including our framework. Our embedding framework consistently outperforms the other two embedding methods, demonstrating significantly lower attack success rates across all text classification and natural language inference tasks. Consequently, RobustEmbed achieves the lowest average attack success rate against all adversarial attack techniques. These findings validate the robustness of our embedding framework and highlight the vulnerabilities of the two state-of-the-art sentence embeddings to various adversarial attacks.

Figure 2 depicts the average number of queries required and the resulting accuracy reduction for a set of 1000 attacks on two fine-tuned sentence embeddings. Green data points represent attacks on the RobustEmbed framework, while red points represent attacks on the USCAL approach (Miao et al., 2021). Connected pairs of points are associated with specific attack techniques. Ideally, a robust sentence embedding should be situated in the top-left region of the diagram, indicating that the attack technique necessitates a larger number of queries to deceive the target model while causing minimal performance degradation. The figure illustrates that, for each attack, RobustEmbed exhibits greater stability compared to the USCAL method. In other words, a larger number of queries is required for RobustEmbed, resulting in a lower accuracy reduction (i.e., better performance) compared to USCAL. This observation holds true for all applied adversarial attacks, indicating the robustness of our framework.

### 5.4 Robust Embeddings

We introduce a new task called Adversarial Semantic Textual Similarity (AdvSTS) to evaluate the resilience of sentence embeddings within our representation framework. AdvSTS uses an efficient adversarial approach, such as TextFooler, to manipulate a pair of input sentences in a way that encourages the target model to produce a regression score that deviates as much as possible from the true score (the ground truth label). Consequently, we create an adversarial STS dataset by converting all benign instances from the original dataset into adversarial examples. Similar to the STS task, AdvSTS employs Pearson's correlation metric to assess the correlation between the predicted similarity scores generated by the target model and the human-annotated similarity scores for the adversar-

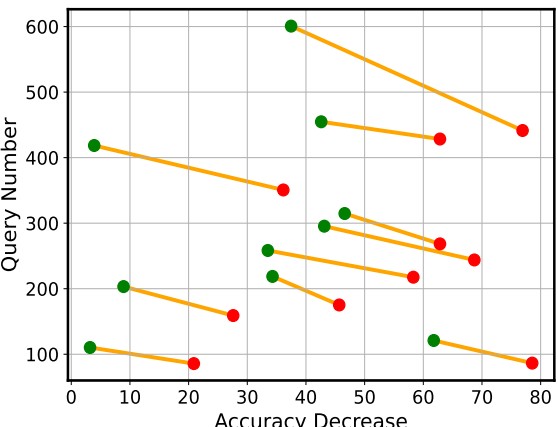

Figure 2: Average number of queries and the resulting accuracy reduction for a set of 1000 attacks on two fine-tuned sentence embeddings. Green points represent attacks on the RobustEmbed framework, while red points represent attacks on the USCAL approach.

ial dataset.

Table 4 illustrates the attack success rates of five different adversarial attack techniques (namely TextFooler, TextBugger, PWWS, BAE, and BERTAttack) applied to three sentence embeddings, including our framework. These evaluations are carried out for two specific AdvSTS tasks, namely AdvSTS-B and AdvSICK-R. Notably, our embedding framework consistently outperforms the other two embedding methods, showing significantly lower attack success rates across both AdvSTS tasks and all employed adversarial attack techniques.

In conclusion, the extensive experiments conducted and the results presented in Tables 1, 2, 3, and 4, as well as Figure 2, provide strong evidence of the exceptional performance of RobustEmbed in various text representation and classification tasks, as well as its resilience against various adversarial attacks and tasks. These findings support the notion that our framework possesses remarkable generalization and robustness capabilities, underscoring its potential as an efficient and versatile approach for generating high-quality sentence embeddings.

### 5.5 Distribution of Sentence Embeddings

We followed the methodology proposed by Wang and Isola (2020) to employ two critical evaluation metrics, termed *alignment* and *uniformity*, to assess the quality of our representations. In the context of positive pairs represented by the distribution $p_{pos}$, *alignment* calculates the anticipated distance

| Adversarial Attack | Model | AdvSTS-B | AdvSICK-R | Avg. |
|---|---|---|---|---|
| TextFooler | SimCSE-BERT$_{base}$ | 21.07 | 24.17 | 22.62 |
| | USCAL-BERT$_{base}$ | 16.52 | 18.71 | 17.62 |
| | RobustEmbed-BERT$_{base}$ | **7.48** | **8.95** | **8.22** |
| TextBugger | SimCSE-BERT$_{base}$ | 27.49 | 28.34 | 27.91 |
| | USCAL-BERT$_{base}$ | 21.52 | 24.88 | 23.20 |
| | RobustEmbed-BERT$_{base}$ | **11.76** | **13.01** | **12.39** |
| PWWS | SimCSE-BERT$_{base}$ | 24.15 | 26.82 | 25.49 |
| | USCAL-BERT$_{base}$ | 21.28 | 23.65 | 22.47 |
| | RobustEmbed-BERT$_{base}$ | **13.56** | **14.44** | **14.00** |
| BAE | SimCSE-BERT$_{base}$ | 26.92 | 28.81 | 27.86 |
| | USCAL-BERT$_{base}$ | 22.92 | 25.48 | 24.20 |
| | RobustEmbed-BERT$_{base}$ | **11.13** | **12.82** | **11.98** |
| BERTAttack | SimCSE-BERT$_{base}$ | 31.60 | 32.85 | 32.23 |
| | USCAL-BERT$_{base}$ | 26.02 | 28.51 | 27.26 |
| | RobustEmbed-BERT$_{base}$ | **12.99** | **13.18** | **13.09** |

Table 4: Attack success rates of five adversarial attack techniques applied to three sentence embeddings (SimCSE, USCAL, and RobustEmbed) across two Adversarial STS (AdvSTS) tasks (i.e. AdvSTS-B and AdvSICK-R).

between the embeddings of paired instances:

$$\ell_{\text{align}} \triangleq \mathop{\mathbb{E}}_{(x,x^+) \sim p_{pos}} \| f(x) - f(x^+) \|^2. \quad (9)$$

*Uniformity* quantifies how uniformly the embeddings are distributed within the representation space:

$$\ell_{\text{uniform}} \triangleq \log \mathop{\mathbb{E}}_{x,y \overset{i.i.d.}{\sim} p_{\text{data}}} e^{-2\| f(x) - f(y) \|^2}, \quad (10)$$

where $p_{data}$ represents the data distribution. The underlying principle of these metrics is that positive instances should remain closely grouped, while embeddings for random instances should be spread across the hypersphere. Figure 3 illustrates the *uniformity* and *alignment* of various sentence embedding models, where lower values correspond to improved performance. In comparison to alternative representations, RobustEmbed achieves a similar level of *uniformity* (-2.293 vs. -2.305) but demonstrates superior *alignment* (0.058 vs. 0.073). This highlights the greater efficiency of our framework in optimizing the representation space in two distinct directions.

## 6 Conclusion and Future Work

In this paper, we proposed RobustEmbed, a self-supervised sentence embedding framework that significantly enhances robustness against various adversarial attacks while achieving state-of-the-art

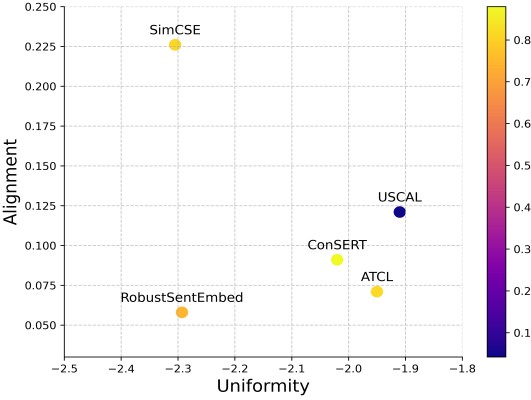

Figure 3: $\ell_{\text{align}} - \ell_{\text{uniform}}$ plot of models based on BERT$_{base}$

performance in a wide range of text representation and NLP tasks. Current sentence embeddings are vulnerable to adversarial attacks. RobustEmbed fills this gap by leveraging high-risk adversarial perturbations within a novel contrastive objective approach. We demonstrated the effectiveness of our framework through extensive experiments on semantic textual similarity and transfer learning tasks. Furthermore, Empirical findings substantiate the robustness of RobustEmbed against diverse adversarial attacks. As future work, we aim to explore the use of hard negative examples in the supervised setting to further enhance the efficiency of text representations.

## Limitations

Despite the ingenuity of our methodology and its impressive performance, our framework does have some potential limitations:

- Our framework is primarily designed and optimized for descriptive models, such as BERT, which excel in understanding and representing language, as well as related tasks like text classification. However, it may not be directly applicable to generative models like GPT, which prioritize generating coherent and contextually relevant text. Therefore, there may be limitations in applying our methodology to enhance the generalization and robustness characteristics of generative pre-trained models.

- Our framework requires significant GPU resources for pre-training large-scale pre-trained models like RoBERTa$_{large}$. Due to limitations in GPU availability, we had to utilize smaller batch sizes during pre-training. While larger batch sizes (e.g., 256 or 512) generally lead to improved performance metrics, our experiments had to compromise and use smaller batch sizes to generate sentence embeddings efficiently given the GPU resource constraints.

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

## A    Training Details

In our experimental setup, we initialize our sentence encoder, denoted as $f_\theta$, using the checkpoints obtained from BERT (Devlin et al., 2019) and RoBERTa (Liu et al., 2019). For sentence embedding, RobustEmbed utilizes the representation of the [CLS] token as the starting point and incorporates a pooler layer on top of the [CLS] representations to facilitate contrastive learning objectives. The training process of RobustEmbed involves 2 epochs, with model evaluation conducted every 250 training steps. The best checkpoint, determined by the highest average STS (Semantic Textual Similarity) score, is selected for final evaluation. To train the model, we utilize a dataset consisting of $10^6$ randomly sampled sentences from English Wikipedia, as provided by the SimCSE framework (Gao et al., 2021). The average training time for RobustEmbed is 2-4 hours. As our framework is initialized with pre-trained checkpoints, it exhibits robustness that is not sensitive to batch sizes, thus enabling us to employ batch sizes of either 64 or 128. In terms of transfer tasks, we determine the best hyperparameters based on the averaged score obtained from the development sets of six transfer tasks.

## B    Ablation Studies

In this section, we analyze the influence of four key hyperparameters in our approach on the overall performance. We utilize BERT$_{base}$ as the encoder and evaluate the hyperparameters using the development set of STS tasks.

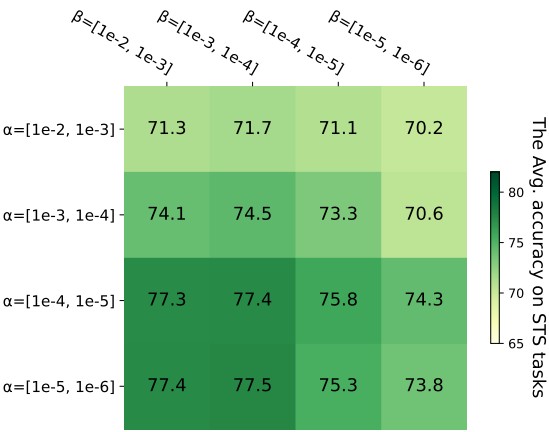

Figure 4: The impact of step sizes in perturbation generation on the average performance of STS tasks.

### B.1    Step Sizes in Perturbation Generation

As depicted in Algorithm 1, the RobustEmbed framework incorporates two step sizes, denoted as $\alpha$ and $\beta$, to perform iterative updates during the PGD and FGSM perturbation generation processes, respectively. Figure 4 illustrates the collaborative effect of varying ranges for these two step sizes in generating high-risk perturbations, which is significant for achieving efficient contrastive learning objective. The results indicate greater improvement when $\beta$ is adjusted in a lower range while $\alpha$ is placed in an upper range. Specifically, better performance is observed when $\alpha$ and $\beta$ are assigned ranges of [1e-4, 1e-6] and [1e-2, 1e-4], respectively. Therefore, we utilize $\alpha$ = 1e-5 and $\beta$ = 1e-3 for our experiments as it achieves the best results among the different arrangements.

### B.2    Step Numbers in Perturbation Generation

RobustEmbed applies T-step FGSM and K-step PGD iterations to obtain high-risk adversarial perturbations for the contrastive learning objective. To simplify the analysis of perturbation generation iterations, we set K = T. Figure 5 demonstrates the impact of different step numbers (N = K or T) on effectiveness. We observe a gradual improvement as N increases from 1 to 9; however, beyond N=9, the improvement becomes negligible. Moreover, a higher N leads to longer running-time and unfair resource allocation. Hence, we select N=5 for our experiments.

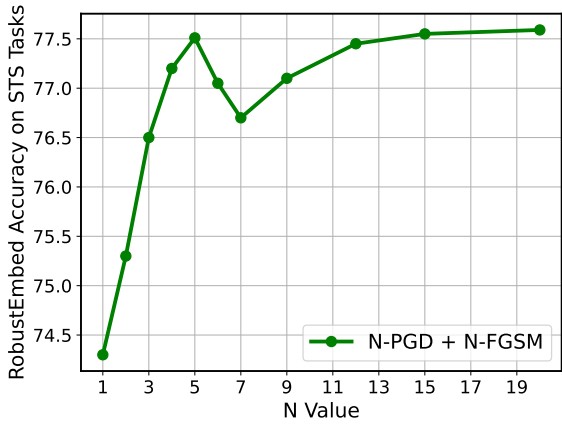

Figure 5: The effect of the step number (denoted as N = K or T) in the T-step FGSM and K-step PGD methods on the averaged correlation of the different Semantic Textual Similarity (STS) tasks.

### B.3 Norm Constraint

To ensure the imperceptibility of the generated adversarial examples, the magnitude of the perturbation vector, denoted as $\delta$, is controlled in RobustEmbed. Three commonly used norm functions, namely $L_1$, $L_2$, and $L_\infty$, are employed to restrict the magnitude of $\delta$ to small values. Table 5 presents the averaged Spearman's correlation of these norm functions across different Semantic Textual Similarity tasks. The $L_\infty$ norm demonstrates superior correlation compared to the other two norms, thus it is selected as the norm function for our experimental evaluation.

| Norm | Correlation |
| --- | --- |
| $L_\infty$ | **77.51** |
| $L2$ | 76.82 |
| $L1$ | 75.28 |

Table 5: The influence of the norm constraint on perturbation generation on the average performance of various Semantic Textual Similarity (STS) tasks.

### B.4 Modulation Factor

RobustEmbed incorporates a modulation factor, denoted as $0 \leq \lambda \leq 1$, to adjust the relative significance of each separate perturbation (PGD and FGSM) in the formation of the final perturbation. The performance efficiency of various values for this modulation factor on semantic textual similarity tasks is presented in Table 6. The results indicate that $\lambda = 0.5$ achieves the highest averaged correlation among the tested magnitudes, indicat-

ing its effectiveness in generating more powerful perturbations. Therefore, we adopt this setting in the configuration of our framework.

| $\lambda$ | Correlation |
| --- | --- |
| 0 | 76.36 |
| 0.25 | 76.91 |
| 0.5 | **77.51** |
| 0.75 | 77.04 |
| 1 | 76.48 |

Table 6: The impact of the modulation factor on the average performance of different Semantic Textual Similarity (STS) tasks in generating the final perturbation.

## C  Adversarial Training Comparison

To compare our framework with other standard adversarial training methods, we fine-tuned our pre-trained model using a similar adversarial training approach as the one employed during the pre-training phase. Subsequently, we compared the fine-tuned model with three standard adversarial training methods, namely PGD, FreeLB (Zhu et al., 2020), and SMART (Jiang et al., 2020), after the fine-tuning step, and presented the experimental results in table 7. As shown, our framework outperforms the three other adversarial training methods, achieving the highest average accuracy for STS and transfer tasks and the lowest average attack success rate under TextFooler, TextBugger, and BERTAttack attacks.

## D  Contrastive Learning Loss

The first part of the total contrastive loss (Equation 8) optimizes the similarity between the input instance $x$ and its positive pair ($x^{pos}$), along with the similarity between $x$ and its adversarial perturbation ($x^{adv}$). Although it indirectly brings $x^{pos}$ and $x^{adv}$ closer, our observations show that regularizing the main objective function (Equation 7) through direct contrastive learning between $x^{pos}$ and $x^{adv}$ (the second part of Equation 8) helps us achieve improved clean accuracy and robustness. Table 8 illustrates the effect of different values of the regularization parameter ($\gamma$) on the final performance of our framework. As can be seen, when $\gamma = 1/128$, the framework achieves the highest average accuracy for STS and transfer tasks and the lowest average attack success rate under the TextFooler attack. We employ $\gamma = 1/128$ for all other experiments.

| Model | STS | Transfer | TextFooler | TextBugger | BERTAttack |
|---|---|---|---|---|---|
| PGD | 76.37 | 79.15 | 50.33 | 31.05 | 49.72 |
| FreeLB | 81.91 | 86.03 | 48.70 | 27.11 | 47.83 |
| SMART | 82.65 | 87.34 | 45.46 | 26.08 | 47.39 |
| RobustEmbed | **85.79** | **89.86** | **37.12** | **20.43** | **39.25** |

Table 7: Performance Comparison of Adversarial Training Methods

| $\gamma$ | STS | Transfer | TextFooler |
|---|---|---|---|
| 1/64 | 76.46 | 85.93 | 44.37 |
| 1/128 | **77.51** | **86.20** | **37.97** |
| 1/256 | 77.06 | 85.87 | 40.32 |
| 1/512 | 75.84 | 84.66 | 42.58 |

Table 8: Effect of Regularization Parameter ($\gamma$) on our Framework Performance

# E    Text Classification Tasks

This section presents additional information on the text classification tasks used to assess the generalization and robustness capabilities of our framework in comparison to various sentence embedding methods. The MR (Movie Reviews) dataset (Pang and Lee, 2005b) consists of sentence-level samples with sentiment polarity, comprising 8,530 training and 1,066 testing highly polar instances. The CR dataset (Hu and Liu, 2004) is a customer review dataset collected in three steps: extracting products with customer comments, identifying opinion sentences, and labeling each sentence as positive or negative. The SUBJ dataset (Pang and Lee, 2004) contains 5,000 subjective and 5,000 objective sentences from movie reviews, labeled based on subjectivity status and polarity. The MPQA dataset (Wiebe et al., 2005) includes annotated documents from diverse news sources, categorizing opinion states such as beliefs, emotions, sentiments, and speculations. The SST2 dataset (Socher et al., 2013) is a sentence-level dataset with 8,544 training and 2,210 testing highly polar samples, extracted from movie reviews and classified as negative or positive. The MRPC dataset (Dolan and Brockett, 2005) contains 5,801 sentence pairs from news articles, labeled by human annotators to indicate semantic equivalence relationships. The YELP Polarity Review (YELP) dataset (Zhang et al., 2015) consists of document-level samples, with 560,000 training and 38,000 testing highly polar instances classified as negative (1- and 2-star) or positive (4- and 5-star) reviews. The Internet Movie Database

(IMDb) Review dataset (Maas et al., 2011) contains 25,000 training and 25,000 testing highly polar samples, with negative and positive classes corresponding to review scores of $\leq 4$ and $\geq 7$ out of 10, respectively. Rotten Tomatoes Movie Reviews (MR) (Pang and Lee, 2005a) is a sentence-level dataset consisting of 8,530 training and 1,066 testing highly polar samples, where negative and positive classes are assigned based on calibration among different critics. SNLI (Bowman et al., 2015) (MNLI (Williams et al., 2018)) is a three-class dataset comprising 550,152 (392,702) training and 10,000 (19,643) testing human-written sentence pairs in English. Each set of three pairs in SNLI (MNLI) is created using a different image caption from the Flicker30K dataset (Young et al., 2014) (ten sources of text), with the premise sentence serving as the first sentence in each set. The hypothesis sentence of the first, second, and third pair is generated to be in entailment (category 1), contradiction (category 2), and neutral (category 3) with the respective premise sentence. While SNLI uses premise sentences from a relatively homogeneous image caption dataset, MNLI covers a broader range of text styles. The MNLI testing sample pairs are divided into two categories: "Matched" and "Mismatched," where MNLI-Matched pairs share similar context and resemblance to the training pairs compared to MNLI-Mismatched pairs.

# F    Adversarial Attack Methods

This section presents additional details on the diverse adversarial attack techniques employed to assess the robustness of our sentence embedding framework. The TextBugger method (Li et al., 2019) identifies important words using the Jacobian matrix of the target model and selects an optimal perturbation from five types of generated perturbations. The PWWS method (Ren et al., 2019) utilizes a synonym-swap technique based on a combination of word saliency scores and maximum word-swap effectiveness. TextFooler (Jin et al., 2020) identifies important words, gathers

synonyms, and replaces each important word with the most semantically similar and grammatically correct synonym. The BAE method (Garg and Ramakrishnan, 2020) employs four adversarial attack strategies involving word replacement or/and word insertion operations, where a portion of the text is masked and BERT MLM is used to generate substitutions. The BERTAttack method (Li et al., 2020b) consists of two steps: (a) searching for vulnerable words/sub-words and (b) using BERT MLM to generate semantic-preserving substitutes for the vulnerable tokens.