# OpenReview forum: "RobustEmbed: Robust Sentence Embeddings Using Self-Supervised Contrastive Pre-Training"
_EMNLP/2023/Conference — EMNLP 2023 Findings_

### Official Review · Reviewer_hADL · 2023-07-26

**Typos Grammar Style And Presentation Improvements:** NA
**Soundness:** 4

**Excitement:**

3: Ambivalent: It has merits (e.g., it reports state-of-the-art results, the idea is nice), but there are key weaknesses (e.g., it describes incremental work), and it can significantly benefit from another round of revision. However, I won't object to accepting it if my co-reviewers champion it.

**Missing References:**

Shi, Jiahui, Linjing Li, and Daniel Zeng. "ASCL: Adversarial supervised contrastive learning for defense against word substitution attacks." Neurocomputing 510 (2022): 59-68.

**Paper Topic And Main Contributions:**

This paper introduces contrastive learning to enhance sentence embeddings of PLMs for improved adversarial robustness. Experiment results show significant robustness improvements on STS and transfer tasks.

**Questions For The Authors:**

NA

**Reasons To Accept:**

* This work tackles the valuable problem of adversarial robustness.

* The structure of the paper is clean.

**Reasons To Reject:**

* Experiments on adversarial robustness with 100 examples are not convincing, even though the author ran them five times. The common practice is to use at least 1000 examples.

* There is a lack of comparison with other basic adversarial training methods, such as basic PGD or FreeLB.

* As a critical point, the reviewer remains unconvinced that solely using unsupervised contrastive loss in perturbation construction can lead to adversarial robustness.

**Reproducibility:**

4: Could mostly reproduce the results, but there may be some variation because of sample variance or minor variations in their interpretation of the protocol or method.

**Reviewer Confidence:**

4: Quite sure. I tried to check the important points carefully. It's unlikely, though conceivable, that I missed something that should affect my ratings.

---

> ### Author Rebuttal · Authors · 2023-08-29
>
> ### Reasons To Reject (Question 1)
>
> Question: Experiments on adversarial robustness with 100 examples are not convincing, even though the author ran them five times. The common practice is to use at least 1000 examples.
>
> #### Response:
>
> To address the concern raised by the reviewer, we conducted adversarial robustness experiments with 1,000 samples per experiment. To ensure statistical validity, each experiment was repeated five times. The table below presents the success rates of three adversarial attack techniques (TextFooler, TextBugger, and BERTAttack) on three sentence embeddings, including our framework, for three downstream tasks (IMDB, MRPC, and SNLI). Our embedding framework outperforms the other three methods, demonstrating remarkably lower attack success rates across three classification tasks and all adversarial attack techniques. We plan to expand our experiments by incorporating additional adversarial attacks and more downstream classification tasks to provide a comprehensive assessment of the robustness in sentence embeddings.
>
> |Adversarial Attack|Model        	         |IMDB        |MRPC     |SNLI         |
> |------------------------|-----------------------------|--------------|-------------|--------------|
> |TextFooler    	      |SimCSE-BERT  	  |72.50       |75.82       |69.42       |
> |TextFooler    	      |USCAL-BERT   	  |59.92       |59.43       |53.17       |
> |TextFooler    	      |RobustEmbed-BERT | **38.13** | **37.09** | **37.25**|
> |TextBugger           |SimCSE-BERT  	  |50.73       |54.91       |46.26       |
> |TextBugger           |USCAL-BERT   	  |38.14       | 37.05      | 31.71      |
> |TextBugger           |RobustEmbed-BERT | **22.52** |**21.62**  |**20.16** |
> |BERTAttack          |SimCSE-BERT  	  |74.66       |79.26       |75.52       |
> |BERTAttack          |USCAL-BERT   	  |61.52       | 60.87	    |54.03       |
> |BERTAttack          |RobustEmbed-BERT |**41.73**  |**40.65** |**41.49**  |
>
> -----------------------------------------------------------------------------------------------------------
> -----------------------------------------------------------------------------------------------------------
>
> ### Reasons To Reject (Question 2)
>
> Question: There is a lack of comparison with other basic adversarial training methods, such as basic PGD or FreeLB.
>
> #### Response:
>
> FreeLB employs supervised adversarial training during fine-tuning, while our framework utilizes self-supervised adversarial training during pre-training. Due to this fundamental difference, a direct fair comparison between these methods is not feasible. Our framework is therefore only comparable with unsupervised/self-supervised adversarial training methods at the pre-training stage. In this context, we have conducted comparative assessments with two state-of-the-art methods, ConSERT [1] and USCAL [2], as detailed in Tables 1-3 of our submitted manuscript. To address the concern raised by the reviewer, we compared our framework with three unsupervised adversarial training methods, namely PGD, Multi-step PGD [3], and ALUM [4], and presented the experimental results in the following table.  As shown, the framework outperforms the three other baselines by achieving the highest average accuracy for STS and transfer tasks and the lowest average attack success rate under TextFooler, TextBugger, and BERTAttack attacks. We plan to expand our experiments by incorporating additional adversarial attacks and also providing task-specific results for all of the STS and transfer tasks utilized in our research work.
>
> |Model                          |STS          |Transfer    |TextFooler |TextBugger  |BERTAttack   |
> |------------------------------|--------------|--------------|---------------|----------------|-------------------|
> |PGD-BERT                   |64.24       |76.15  	|51.36         |30.45           |49.17             |
> |Multi-step PGD-BERT  |66.91       |80.81       |47.14         |28.09           |46.96             |
> |ALUM-BERT       	      |68.25       |84.37  	|46.70         |27.38   	|46.38             |
> |RobustEmbed-BERT    |**77.51** |**86.20**  |**37.97**   |**20.98**      |**39.62**       |
>
>
> ### References
> 1. Yuanmeng Yan, Rumei Li, Sirui Wang, Fuzheng Zhang, Wei Wu, and Weiran Xu. (2021). ConSERT: A Contrastive Framework for Self-Supervised Sentence Representation Transfer. In _Proceedings of the 59th Annual Meeting of the Association for Computational Linguistics and the 11th International Joint Conference on Natural Language Processing (pp 5065–5075). Association for Computational Linguistics.
> 2. Deshui Miao, Jiaqi Zhang, Wenbo Xie, Jian Song, Xin Li, Lijuan Jia, and Ning Guo. (2021).  Simple contrastive representation adversarial learning for nlp tasks.
> 3. Chen Zhu, Yu Cheng, Zhe Gan, Siqi Sun, Tom Goldstein, and Jingjing Liu. (2020). FreeLB: Enhanced Adversarial Training for Language Understanding. In Proceedings of the International Conference on Learning Representations (ICLR).
> 4. Liu, X., Cheng, H., He, P., Chen, W., Wang, Y., Poon, H., & Gao, J. (2020). Adversarial training for large neural language models. _arXiv preprint arXiv:2004.08994.
>
> -----------------------------------------------------------------------------------------------------------
> -----------------------------------------------------------------------------------------------------------
>
> ### Reasons To Reject (Question 3)
>
> Question: As a critical point, the reviewer remains unconvinced that solely using unsupervised contrastive loss in perturbation construction can lead to adversarial robustness.
>
> #### Response:
>
> It is noteworthy that unsupervised contrastive loss encourages the model to learn more discriminative features for input data. It encourages the model to push embeddings of similar instances closer together in the feature space while pushing embeddings of dissimilar instances apart. Furthermore, our framework employs an efficient combination of FGSM and PGD techniques iteratively to generate the worst-case perturbation in the embedding space. Subsequently, this perturbation is employed to optimize the similarity between the benign input sample and its corresponding adversarial perturbation in a contrastive manner. Consequently, unsupervised contrastive loss trains the model to align input samples and their worst-case perturbed embeddings within a meaningful continuous space. This strategic alignment effectively encompasses a wide range of adversarial examples in the raw text space. This enhanced discriminative ability makes it more challenging for adversaries to discover perturbations capable of deceiving the model.
>
> Our empirical findings in Table 3 demonstrate the robustness superiority of this idea, employed in our framework, against five challenging adversarial attacks when compared to two state-of-the-art sentence embedding methods.
>
> -----------------------------------------------------------------------------------------------------------
> -----------------------------------------------------------------------------------------------------------
>
> ### Missing References:
> Shi, Jiahui, Linjing Li, and Daniel Zeng. "ASCL: Adversarial supervised contrastive learning for defense against word substitution attacks." Neurocomputing 510 (2022): 59-68.
>
>
> #### Response:
>
> Thank you for noting the missing reference. It is pertinent, and we will include it. Unlike the referenced paper, our framework utilizes high-risk perturbations in a self-supervised contrastive learning approach to enhance the target model's resilience against a range of adversarial attacks.
>
> -----------------------------------------------------------------------------------------------------------
> -----------------------------------------------------------------------------------------------------------
>
> ### Reproducibility: 2
> Would be hard pressed to reproduce the results. The contribution depends on data that are simply not available outside the author's institution or consortium; not enough details are provided.
>
> #### Response:
>
> Our source code and necessary data for training the framework are publicly available in the following GitHub project: (https://github.com/jasl1/RobustEmbed). To train the framework, we utilize a dataset consisting of $10^6$ randomly sampled sentences from the English Wikipedia, as provided by the SimCSE framework [1]. Users can easily reproduce the experimental results reported in the paper by following the instructions available in the README file of the GitHub project. Detailed information about the training configuration is also provided in the "Training Details" (Appendix A) section of our manuscript.
>
>
> ### References
> 1. Tianyu Gao, Xingcheng Yao, and Danqi Chen. (2021). SimCSE: Simple Contrastive Learning of Sentence Embeddings. In _Proceedings of the 2021 Conference on Empirical Methods in Natural Language Processing_ (pp. 6894–6910). Association for Computational Linguistics.

---

### Official Review · Reviewer_ZN7y · 2023-08-05

**Typos Grammar Style And Presentation Improvements:** 1. There are multiple errors for cita…
**Soundness:** 3

**Excitement:**

2: Mediocre: This paper makes marginal contributions (vs non-contemporaneous work), so I would rather not see it in the conference.

**Missing References:**

Recent SOTA methods on self supervised sentence embeddings should be cited.

**Paper Topic And Main Contributions:**

This paper works on robust sentence embeddings with self-supervised learning methods. The authors propose a robust sentence embedding method called RobustEmbed, which uses adversarial perturbations to strengthen sentence embeddings. The proposed method is evaluated on three settings, including semantic textual similarity tasks, six transfer tasks, and text classification tasks. Experimental results show that RobustEmbed can improve over investigated baselines.

**Questions For The Authors:**

Please see Point 4 in Weaknesses.

**Reasons To Accept:**

1. The overall idea is easy to follow and the related implementation will be released.
2. On the classification tasks, the proposed method significantly outperforms the baselines.

**Reasons To Reject:**

1. The improvements on the semantic textual similarity and transfer tasks seem to be marginal.


2. Strong SOTA methods on the unsupervised embedding methods are not included. Another main baseline is USCAL, which was published in 2021. More recent baselines are necessary.


3. The novelty is somewhat limited. The adversarial methods used in this paper including FGSM and PGD are well-known and general methods for embedding perturbation. If there are specific designs which are important to sentence embeddings, the proposed method can be more novel. For example, since embeddings are continuous and not all the embeddings in the continuous space are meaningful, how to ensure the adversarial perturbations are semantically meaningful can be worth thinking about.


4. Another major problem lies in the evaluation. The current experiment suggests the robustness of the proposed method using text classification tasks which are originally designed for detecting the robustness or BERT. But does robustness in classification mean robustness in sentence embeddings? Or does robustness in BERT mean robustness in sentence embeddings? The evaluation set are not strongly related to sentence embeddings. For example, two paraphrase sentences “this movie is not bad at all” vs. “This film is actually quite good.”, their semantics are almost the same, so the sentence embeddings should be very close. But for classification performances, different models might give different results. Similarly, when the classification results for two sentences are the same do not mean that their sentence embeddings are close to each other. To solve these problems, a new evaluation benchmark is necessary, which is specially designed for direct evaluations of sentence embeddings.

**Reproducibility:**

4: Could mostly reproduce the results, but there may be some variation because of sample variance or minor variations in their interpretation of the protocol or method.

**Reviewer Confidence:**

5: Positive that my evaluation is correct. I read the paper very carefully and I am very familiar with related work.

---

> ### Author Rebuttal · Authors · 2023-08-29
>
> ### Reasons To Reject (Question 1)
>
> Question: The improvements on the semantic textual similarity and transfer tasks seem to be marginal.
>
> #### Response:
>
> Recent research has introduced sentence embeddings based on Pre-trained Language Models (PLMs), demonstrating promising performance across a range of semantic textual similarity and various downstream classification tasks. However, these embeddings exhibit vulnerability in adversarial environments, especially when subjected to external attacks. The core concept behind our proposed framework is to effectively address this vulnerability by generating robust sentence embeddings that are resilient in adversarial settings, particularly against a variety of adversarial attacks, while still maintaining proficiency in generalization, as evidenced by their promising performance across diverse downstream tasks.
>
> The extensive experiments we have conducted demonstrate the exceptional performance of the framework in defending against state-of-the-art adversarial attacks within adversarial environments, in addition to showcasing improvements in semantic textual similarity and transfer tasks. The collective experimental results from Tables 1 to 3 substantiate the framework's capacity to successfully achieve its primary objectives while also enhancing performance in various semantic textual similarity and downstream classification tasks.
>
> -----------------------------------------------------------------------------------------------------------
> -----------------------------------------------------------------------------------------------------------
>
> ### Reasons To Reject (Question 2)
>
> Question: Strong SOTA methods on the unsupervised embedding methods are not included. Another main baseline is USCAL, which was published in 2021. More recent baselines are necessary.
>
> #### Response:
>
> We have already compared our framework with four recent baselines, including the USCAL baseline: SimCSE (2021) [1], ConSERT (2021) [2], USCAL (2021) [3], and ATCL (2022) [4]. The framework outperforms the USCAL embeddings in all conducted experiments by achieving higher average accuracy for STS and transfer tasks and a lower average attack success rate in the adversarial setting (Tables 1-3 in our submitted manuscript). To address the concern raised by the reviewer, we compared our framework with two other recent baselines, namely ESimCSE [5] and DiffCSE [6], and presented the experimental results in the following table. All experimental results were reproduced and reevaluated using our own configuration. As shown, the framework outperforms the two other embedding baselines by achieving the highest average accuracy for STS and transfer tasks and the lowest average attack success rate under TextFooler, TextBugger, and BERTAttack attacks. We will expand our experiments by incorporating additional adversarial attacks and also providing task-specific results for all of the STS and transfer tasks utilized in our work.
>
> |Model                         |STS       |Transfer   |TextFooler  |TextBugger |BERTAttack |
> |-----------------------------|------------|-------------|---------------|----------------|-----------------|
> |ESimCSE-BERT        |76.29	   |85.90       |65.19         |41.77          |64.19           |
> |DiffCSE-BERT      	   |76.40     |85.74       |54.91         |36.72          |52.33           |
> |RobustEmbed-BERT |**77.51**|**86.20**  |**37.97**   |**20.98**    |**39.62**      |
>
> ### References
> 1. Tianyu Gao, Xingcheng Yao, and Danqi Chen. (2021). SimCSE: Simple Contrastive Learning of Sentence Embeddings. In _Proceedings of the 2021 Conference on Empirical Methods in Natural Language Processing_ (pp. 6894–6910). Association for Computational Linguistics.
> 2.  Yuanmeng Yan, Rumei Li, Sirui Wang, Fuzheng Zhang, Wei Wu, and Weiran Xu. (2021). ConSERT: A Contrastive Framework for Self-Supervised Sentence Representation Transfer. In _Proceedings of the 59th Annual Meeting of the Association for Computational Linguistics and the 11th International Joint Conference on Natural Language Processing (pp 5065–5075). Association for Computational Linguistics.
> 3. Deshui Miao, Jiaqi Zhang, Wenbo Xie, Jian Song, Xin Li, Lijuan Jia, and Ning Guo. (2021).  Simple contrastive representation adversarial learning for nlp tasks.
> 4. Daniela N. Rima, DongNyeong Heo, and Heeyoul Choi. Adversarial training with contrastive learning in nlp.  Computer Speech & Language. Submitted.
> 5.  Xing Wu, Chaochen Gao, Liangjun Zang, Jizhong Han, Zhongyuan Wang, and Songlin Hu. (2022). ESimCSE: Enhanced Sample Building Method for Contrastive Learning of Unsupervised Sentence Embedding. In _Proceedings of the 29th International Conference on Computational Linguistics_ (pp. 3898–3907). International Committee on Computational Linguistics.
> 6. Yung-Sung Chuang, Rumen Dangovski, Hongyin Luo, Yang Zhang, Shiyu Chang, Marin Soljacic, Shang-Wen Li, Scott Yih, Yoon Kim, and James Glass. (2022). DiffCSE: Difference-based Contrastive Learning for Sentence Embeddings. In _Proceedings of the 2022 Conference of the North American Chapter of the Association for Computational Linguistics: Human Language Technologies_ (pp. 4207–4218). Association for Computational Linguistics.
> -----------------------------------------------------------------------------------------------------------
> -----------------------------------------------------------------------------------------------------------
>
> ### Reasons To Reject (Question 3)
>
> Question: The novelty is somewhat limited. The adversarial methods used in this paper including FGSM and PGD are well-known and general methods for embedding perturbation. If there are specific designs which are important to sentence embeddings, the proposed method can be more novel. For example, since embeddings are continuous and not all the embeddings in the continuous space are meaningful, how to ensure the adversarial perturbations are semantically meaningful can be worth thinking about.
>
> #### Response:
>
> In recent research, various adversarial techniques, including Fast Gradient Sign Method (FGSM), Fast Gradient Method (FGM), Projected Gradient Descent (PGD), and Multi-step PGD (K-PGD), have been employed to generate adversarial perturbations in continuous embedding spaces. These techniques, although not originally designed for sentence embeddings, have been adapted for consistency within this space. Our framework intelligently expands upon the FGSM by introducing a Multi-step FGSM approach (T-FGSM) to acquire the perturbation that represents the worst-case scenario. As both K-PGD and T-FGSM are applied within the same embedding context, they can be combined into a novel practical perturbation generator. Consequently, the RobustEmbed framework employs a parameter $\lambda$ to balance the relative significance of each individual perturbation in generating the final perturbation (Equation 5).
>
> We employ two key metrics [1] to assess the semantic meaningfulness of our sentence embedding framework, namely, alignment and uniformity. The metrics rely on the idea that similar semantic instances should be close together, while embeddings for random instances should be spread widely across the hypersphere. In our experimental results, which are shown in the following table, we evaluate the alignment and uniformity metrics for our framework using various perturbation generation techniques during training. Smaller metric values indicate better performance. Notably, in comparison to the other two perturbation generation methods, the combination of K-PGD and T-FGSM (K-PGD + T-FGSM) demonstrates superior efficiency in generating semantically meaningful perturbations within the embedding space.
>
> |Perturbation                   |Alignment  |Uniformity  |STS           |
> |--------------------------------|---------------|---------------|---------------|
> |_FGSM_                        |0.258         |  -1.005      |74.95         |
> |_PGD_                          |0.172          |-1.653        |76.16         |
> |_K-PGD_ + _T-FGSM_ |**0.071**    |**-2.295**   |  **77.51** |
>
>
> ### References
> 1. Wang, T., & Isola, P. (2020, November). Understanding contrastive representation learning through alignment and uniformity on the hypersphere. In _International Conference on Machine Learning_ (pp. 9929-9939). PMLR.
>
> ------------------------------------------------------------------------------------------------------------------------------------------------------------------------------
> ------------------------------------------------------------------------------------------------------------------------------------------------------------------------------
>
> ### Reasons To Reject (Question 4)
>
> Question: Another major problem lies in the evaluation. The current experiment suggests the robustness of the proposed method using text classification tasks which are originally designed for detecting the robustness or BERT. But does robustness in classification mean robustness in sentence embeddings? Or does robustness in BERT mean robustness in sentence embeddings? The evaluation set are not strongly related to sentence embeddings. For example, two paraphrase sentences “this movie is not bad at all” vs. “This film is actually quite good.”, their semantics are almost the same, so the sentence embeddings should be very close. But for classification performances, different models might give different results. Similarly, when the classification results for two sentences are the same do not mean that their sentence embeddings are close to each other. To solve these problems, a new evaluation benchmark is necessary, which is specially designed for direct evaluations of sentence embeddings.
>
> #### Response:
>
> Semantic Textual Similarity (STS) is an NLP task that measures the similarity between text segments, usually sentences, by converting them into numerical vectors using sentence embeddings. This allows for comparing sentences based on their underlying semantic content to determine how closely their meanings match in a continuous vector space. To address the concern raised by the reviewer, we introduced a new task called Adversarial Semantic Textual Similarity (AdvSTS) to evaluate the robustness of sentence embeddings in our representation framework. AdvSTS employs an efficient adversarial technique, like TextFooler, to manipulate an input sentence pair in a way that misleads the target model into generating a regression score that maximally deviates from the actual score (truth label). Consequently, we create an adversarial STS dataset by transforming all benign instances from the original dataset into adversarial examples. Similar to the STS task, AdvSTS uses Pearson's correlation metric to assess the correlation between the predicted similarity scores produced by the target model and the human-annotated similarity scores for the adversarial dataset.
>
> The table below presents the success rates of three adversarial attack techniques (TextFooler, TextBugger, and BERTAttack) on three sentence embeddings, including our framework, for two AdvSTS tasks (AdvSTS-B and AdvSICK-R). Our embedding framework outperforms the other two methods, demonstrating significantly lower attack success rates across both AdvSTS tasks and all adversarial attack techniques. We will expand our experiments by incorporating additional adversarial attacks and potentially more embedding methods to provide a comprehensive assessment of the robustness in sentence embeddings.
>
>
> |Adversarial Attack    |Model       |AdvSTS-B   |AdvSICK-R    |Avg.        |
> |---------------------------|--------------|-----------------|------------------|--------------|
> |TextFooler         |SimCSE-BERT             |21.07          |24.17          |22.62        |
>  |TextFooler        |USCAL-BERT               |16.52          |18.71          |71.61        |
> |TextFooler         |RobustEmbed-BERT   | **7.48**      | **8.95**     | **8.22**    |
> |TextBugger        |SimCSE-BERT            |27.49          |28.34          |27.91        |
> |TextBugger        |USCAL-BERT              |21.52          | 24.88         | 23.2         |
> |TextBugger        |RobustEmbed-BERT  | **11.76**    |**13.01**    |**12.39**   |
> |BERTAttack      |SimCSE-BERT            |31.60           |32.85          |32.23        |
> |BERTAttack      |USCAL-BERT              |26.02           | 28.51         |27.26        |
> |BERTAttack      |RobustEmbed-BERT  |**12.99**      |**13.18**    |**13.09**   |
>
> ------------------------------------------------------------------------------------------------------------------------------------------------------------------------------
> ------------------------------------------------------------------------------------------------------------------------------------------------------------------------------
>
> ### Missing References:
> Recent SOTA methods on self supervised sentence embeddings should be cited.
>
> #### Response:
>
> Thank you for pointing out the missing references. We will include the following state-of-the-art (SOTA) self-supervised sentence embeddings in our paper:
>
> ### References
>
> 1.  Xing Wu, Chaochen Gao, Liangjun Zang, Jizhong Han, Zhongyuan Wang, and Songlin Hu. (2022). ESimCSE: Enhanced Sample Building Method for Contrastive Learning of Unsupervised Sentence Embedding. In _Proceedings of the 29th International Conference on Computational Linguistics_ (pp. 3898–3907). International Committee on Computational Linguistics.
> 2. Yung-Sung Chuang, Rumen Dangovski, Hongyin Luo, Yang Zhang, Shiyu Chang, Marin Soljacic, Shang-Wen Li, Scott Yih, Yoon Kim, and James Glass. (2022). DiffCSE: Difference-based Contrastive Learning for Sentence Embeddings. In _Proceedings of the 2022 Conference of the North American Chapter of the Association for Computational Linguistics: Human Language Technologies_ (pp. 4207–4218). Association for Computational Linguistics.
>
> ------------------------------------------------------------------------------------------------------------------------------------------------------------------------------
> ------------------------------------------------------------------------------------------------------------------------------------------------------------------------------
>
> ### Typos Grammar Style And Presentation Improvements:
> There are multiple errors for citation format. For example, “SimCSE Gao et al. (2021) introduced”, “Pan et al. Pan et al. (2022)”. Try to use cite or citet (or newcite) in the correct way.
>
> #### Response:
>
> Thank you for bringing the citation errors to our attention. We will rectify these errors in the final version of the paper.
>
> ------------------------------------------------------------------------------------------------------------------------------------------------------------------------------
> ------------------------------------------------------------------------------------------------------------------------------------------------------------------------------
>
> ### Typos Grammar Style And Presentation Improvements:
> Section 3.1 might not be necessary.
>
> Response:
>
> We will remove this section in the final version of the paper.

---

### Official Review · Reviewer_a5gT · 2023-08-05

**Soundness:** 4

**Excitement:**

3: Ambivalent: It has merits (e.g., it reports state-of-the-art results, the idea is nice), but there are key weaknesses (e.g., it describes incremental work), and it can significantly benefit from another round of revision. However, I won't object to accepting it if my co-reviewers champion it.

**Missing References:**

1.	The idea of creating hard positives for sentence embeddings is not new. The authors might want to involve more discussions of this idea in the related work section.[1]

[1]. Self-Supervised Contrastive Learning with Adversarial Perturbations for Defending Word Substitution-based Attacks. In Findings of NAACL 2022.


**Paper Topic And Main Contributions:**

This paper proposes a new contrastive learning method for learning robust sentence embeddings. Specifically, the method first generates two different versions of embeddings of a sentence by applying dropout twice with a backbone encoder.

A perturbation is then added to one of two embeddings to maximize the contrastive learning loss, and thus creates a hard positive for the contrastive learning objective. To generate the perturbation, the authors leverage PGD and FGSM, which move the sentence embedding towards the direction of increasing the contrastive learning objective. PGD and FGSM are performed for K and T steps, respectively. The final perturbation is a balanced combination of the PGD perturbation and the FGSM perturbation.

The learning objective has two terms, with the first one aiming to maximize the similarity between the original sentence embedding, the dropout obtained sentence embedding, and the perturbed sentence embedding. The second term aims to converge the sentence embeddings of the perturbed sentence embedding and the dropout obtained sentence embedding.

Experimental results show the proposed method 1) learns better sentence embeddings for downstream tasks, including sentence similarity and sentence classification tasks, and 2) learns more robust sentence embeddings against several adversarial attacks.


**Reasons To Accept:**

1.	The paper is very well written and easy to follow.
2.	The proposed method is very straightforward and simple. It basically aims to maximize the similarity between the two dropout obtained sentence embeddings and another perturbed sentence embeddings.
3.	Experimental results are thorough, showing gains on both sentence similarity/sentence classification and robustness evaluation tasks.


**Reasons To Reject:**

1.	If I did not miss something, I am a bit confused on how the authors frame PGD and FGSM in the paper. Formally FGSM is a one-step attack without constraint, but in the paper FGSM is basically equivalent to PGD with P = 2. The authors should better clarify on the name conventions.
2.	Lack of more intuitive understanding of the experimental results. The authors should include more evaluations on the sentence embeddings. For example, measuring the uniformity and alignment of the sentence embeddings by evaluating the average cosine or Euclidean similarities between the sentence embeddings.
3.	Not clear to me why the convergence loss in Equation 8 is important, since the x^adv and x^pos are already being encouraged to similar in the first term. The authors should either provide more explanations or include an ablation study.


**Reproducibility:**

4: Could mostly reproduce the results, but there may be some variation because of sample variance or minor variations in their interpretation of the protocol or method.

**Reviewer Confidence:**

4: Quite sure. I tried to check the important points carefully. It's unlikely, though conceivable, that I missed something that should affect my ratings.

---

> ### Author Rebuttal · Authors · 2023-08-29
>
> ### Reasons To Reject (Question 1)
>
> Question: If I did not miss something, I am a bit confused on how the authors frame PGD and FGSM in the paper. Formally FGSM is a one-step attack without constraint, but in the paper FGSM is basically equivalent to PGD with P = 2. The authors should better clarify on the name conventions.
>
> #### Response:
>
> We build upon the FGSM by introducing a perturbation constraint and updating the perturbation iteratively to create the Multi-step FGSM approach (T-FGSM). Essentially, T-step FGSM and K-step PGD are mathematically equivalent when P is either 2 or $\infty$. Their primary distinctions lie in the number of iterations (i.e. T and K) and the step size of the attack (i.e. $\alpha$ and $\beta$) used to modify the input perturbation, ultimately generating a unique high-level perturbation. Since both K-PGD and T-FGSM are employed within the same embedding context, our framework combines these consistent perturbations in a straightforward yet efficient manner to generate the final perturbation for the contrastive learning objective. We appreciate your observation and will address this in greater detail in the final version of the paper.
>
> --------------------------------------------------------------------------------------------------------------------------------------------------------------------------------------
> --------------------------------------------------------------------------------------------------------------------------------------------------------------------------------------
>
> ### Reasons To Reject (Question 2)
> Question: Lack of more intuitive understanding of the experimental results. The authors should include more evaluations on the sentence embeddings. For example, measuring the uniformity and alignment of the sentence embeddings by evaluating the average cosine or Euclidean similarities between the sentence embeddings.
>
> #### Response:
>
> In response to the reviewer's concern, we followed the approach outlined by Wang and Isola [1] to utilize two crucial metrics, alignment and uniformity, for assessing the quality of our representation space. The fundamental concept underpinning these metrics is that positive instances should remain closely situated, while embeddings for random instances should be dispersed across the hypersphere. The subsequent table presents the experimental results of alignment and uniformity metrics for our framework, RobustEmbed, along with two other state-of-the-art embeddings, namely, DiffCSE [2] and SimCSE [3], on the STS-B test set. Smaller values indicate better performance. In comparison to the other two representations, RobustEmbed achieves a similar level of uniformity (-2.295 vs. -2.305) but exhibits superior alignment (0.071 vs. 0.091). This demonstrates that our framework is more efficient in optimizing the representation space in two distinct directions.
>
> |Model               |Alignment  | Uniformity |STS             |
> |---------------------|---------------|---------------|-----------------|
> |SimCSE           |0.226         |**-2.305**   |76.31   	    |
> |DiffCSE            |0.091         |-1.637        |76.18           |
> |RobustEmbed  |**0.071**   |**-2.295**   |  **77.51**   |
>
>
> ### References
>
> 1. Wang, T., & Isola, P. (2020, November). Understanding contrastive representation learning through alignment and uniformity on the hypersphere. In _International Conference on Machine Learning_ (pp. 9929-9939). PMLR.
> 2. Tianyu Gao, Xingcheng Yao, and Danqi Chen. (2021). SimCSE: Simple Contrastive Learning of Sentence Embeddings. In _Proceedings of the 2021 Conference on Empirical Methods in Natural Language Processing_ (pp. 6894–6910). Association for Computational Linguistics.
> 3. Yung-Sung Chuang, Rumen Dangovski, Hongyin Luo, Yang Zhang, Shiyu Chang, Marin Soljacic, Shang-Wen Li, Scott Yih, Yoon Kim, and James Glass. (2022). DiffCSE: Difference-based Contrastive Learning for Sentence Embeddings. In _Proceedings of the 2022 Conference of the North American Chapter of the Association for Computational Linguistics: Human Language Technologies_ (pp.  4207–4218). Association for Computational Linguistics.
>
> --------------------------------------------------------------------------------------------------------------------------------------------------------------------------------------
> --------------------------------------------------------------------------------------------------------------------------------------------------------------------------------------
>
> ### Reasons To Reject (Question 3)
> Question: Not clear to me why the convergence loss in Equation 8 is important, since the x^adv and x^pos are already being encouraged to similar in the first term. The authors should either provide more explanations or include an ablation study.
>
> #### Response:
>
> The first part of the total contrastive loss (Equation 8) optimizes the similarity between the input instance _x_ and its positive pair ($x^{pos}$), along with the similarity between _x_ and its adversarial perturbation ($x^{adv}$). Although it indirectly brings $x^{pos}$ and $x^{adv}$ closer, our observations show that regularizing the main objective function (Equation 7) through direct contrastive learning between $x^{pos}$ and $x^{adv}$ (the second part of Equation 8) helps us achieve improved clean accuracy and robustness. The following table illustrates the effect of different values of the regularization parameter ($\gamma$) on the final performance of our framework. As can be seen, when $\gamma=1/128$, the framework achieves the highest average accuracy for STS and transfer tasks and the lowest average attack success rate under the TextFooler attack. We employ $\gamma=1/128$ for all other experiments.
>
> |$\gamma$ |STS          |Transfer   |TextFooler  |
> |---------------|--------------|--------------|---------------|
> |1/64           |76.46        |85.93       |44.37         |
> |1/128         |**77.51**  |**86.20**  |**37.97**   |
> |1/256         |77.06        |85.87       |40.32         |
> |1/512         |75.84        |84.66       |42.58         |
>
> -------------------------------------------------------------------------------------------------------------------------------------------------------
> -------------------------------------------------------------------------------------------------------------------------------------------------------
>
> ### Missing References:
> The idea of creating hard positives for sentence embeddings is not new. The authors might want to involve more discussions of this idea in the related work section.[1]
>
> [1]. Self-Supervised Contrastive Learning with Adversarial Perturbations for Defending Word Substitution-based Attacks. In Findings of NAACL 2022.
>
> #### Response:
>
> Thank you for noting the missing reference. It is pertinent, and we will include it. Unlike the referenced paper, our framework utilizes high-risk perturbations in a novel self-supervised contrastive learning approach to enhance the target model's resilience against a range of adversarial attacks. The target model is not restricted to the BERT model, and our framework is compatible with other discriminative Pre-trained Language Models (PLMs).

---

### Meta-Review · Area_Chair_4nKD · 2023-09-24

**Recommendation:** 4

**Metareview:**

This paper introduces RobustEmbed, a self-supervised method for learning robust sentence embeddings. The method applies dropout and adversarial perturbations to generate hard positives for contrastive learning. The method optimizes a loss function that encourages similarity between the original, dropout, and perturbed embeddings, while diverging the dropout and perturbed embeddings. The method shows improvement over baselines on various tasks as well, such as semantic textual similarity, transfer learning, and text classification. The paper is well written with a defined structure and provides high resilience against various adversarial attacks being comparable to other embeddings in representation and classification tasks at the same time.

---

### Decision · Program_Chairs · 2023-10-07

**Decision:**

Accept-Findings

**Comment:**

This paper introduces RobustEmbed, a self-supervised method for learning robust sentence embeddings. The method applies dropout and adversarial perturbations to generate hard positives for contrastive learning. The method optimizes a loss function that encourages similarity between the original, dropout, and perturbed embeddings, while diverging the dropout and perturbed embeddings. The method shows improvement over baselines on various tasks as well, such as semantic textual similarity, transfer learning, and text classification. The paper is well written with a defined structure and provides high resilience against various adversarial attacks being comparable to other embeddings in representation and classification tasks at the same time.